# Screening of Antibiotic and Virulence Genes from Whole Genome Sequenced *Cronobacter sakazakii* Isolated from Food and Milk-Producing Environments

**DOI:** 10.3390/antibiotics12050851

**Published:** 2023-05-05

**Authors:** Ondrej Holý, Julio Parra-Flores, Jaroslav Bzdil, Adriana Cabal-Rosel, Beatriz Daza-Prieto, Ariadnna Cruz-Córdova, Juan Xicohtencatl-Cortes, Ricardo Rodríguez-Martínez, Sergio Acuña, Stephen Forsythe, Werner Ruppitsch

**Affiliations:** 1Science and Research Center, Faculty of Health Sciences, Palacký University Olomouc, 77515 Olomouc, Czech Republic; holy.ondrej@seznam.cz; 2Department of Nutrition and Public Health, Universidad del Bío-Bío, Chillán 3800708, Chile; 3Ptacy s.r.o., Valasska Bystrice 194, 75627 Valasska Bystrice, Czech Republic; vetmed@seznam.cz; 4Austrian Agency for Health and Food Safety, Institute for Medical Microbiology and Hygiene, 1220 Vienna, Austria; adriana.cabal-rosel@ages.at (A.C.-R.); beatriz.daza-prieto@ages.at (B.D.-P.);; 5Intestinal Bacteriology Research Laboratory, Hospital Infantil de México Federico Gómez, Mexico City 06720, Mexico; juanxico@yahoo.com (J.X.-C.);; 6Department of Food Engineering, Universidad del Bío-Bío, Chillán 3800708, Chile; sacuna@ubiobio.cl; 7FoodMicrobe.com Ltd., Adams Hill, Keyworth, Nottinghamshire NG12 5GY, UK; sforsythe4j@gmail.com

**Keywords:** antibiotics resistance, virulence genes, *Cronobacter sakazakii*, whole genome sequencing, environment, food

## Abstract

The objective of this study was to use whole-genome sequencing (WGS) to screen for genes encoding for antibiotic resistance, fitness and virulence in *Cronobacter sakazakii* strains that had been isolated from food and powdered-milk-producing environments. Virulence (VGs) and antibiotic-resistance genes (ARGs) were detected with the Comprehensive Antibiotic Resistance Database (CARD) platform, ResFinder and PlasmidFinder tools. Susceptibility testing was performed using disk diffusion. Fifteen presumptive strains of *Cronobacter* spp. were identified by MALDI-TOF MS and ribosomal-MLST. Nine *C. sakazakii* strains were found in the meningitic pathovar ST4: two were ST83 and one was ST1. The *C. sakazakii* ST4 strains were further distinguished using core genome MLST based on 3678 loci. Almost all (93%) strains were resistant to cephalotin and 33% were resistant to ampicillin. In addition, 20 ARGs, mainly involved in regulatory and efflux antibiotics, were detected. Ninety-nine VGs were detected that encoded for OmpA, siderophores and genes involved in metabolism and stress. The IncFIB (pCTU3) plasmid was detected, and the prevalent mobile genetic elements (MGEs) were ISEsa1, ISEc52 and ISEhe3. The *C. sakazakii* isolates analyzed in this study harbored ARGs and VGs, which could have contributed to their persistence in powdered-milk-producing environments, and increase the risk of infection in susceptible population groups.

## 1. Introduction

*Cronobacter* is a genus of enteropathogenic bacteria consisting of seven species: *Cronobacter sakazakii, C. malonaticus, C. universalis, C. turicensis, C. muytjensii, C. dublinensis* and *C. condimenti* [1,2,3]. *C. sakazakii* and *C. malonaticus* are the species with the greatest clinical significance as they have been detected in both individual cases as well as in outbreaks [4].

*C. sakazakii* mainly affects immunocompromised groups ranging from premature newborns infants to older adults [5,6,7]. In infants, the serious symptoms include life-threatening meningitis, septicemia and necrotizing enterocolitis (EN), while in adults, urinary tract infections are more common. The mortality rate associated with the infection of this pathogen is between 15 and 80%, whereas, for cases with neonatal meningitis and septicemia, rates of 15 and 25%, respectively, are seen [8]. Neonatal meningitic cases have been associated with *C. sakazakii* sequence type 4 (ST4) infections [5]. Infant infections are mainly associated with the consumption of contaminated rehydrated powdered infant formula (PIF) for a target age of less than 6 months, although infection through expressed milk can also occur.

*Cronobacter* spp. can be isolated from other foods, such as infant cereals, milk replacers, water, and vegetables, as well as on food preparation surfaces, dairy plant equipment, and the hospital environment [9,10,11,12]. The organism has been isolated from infant formula manufacturing plants around the world and its prevalence ranges from 3 to 30% [4,13,14].

The *Cronobacter* genus underwent wide diversification during its evolution, with *C. sakazakii* and *C. malonaticus* being more pathogenic, while others have a less frequently reported impact on human health [4]. Information on the diversity, pathogenicity and virulence of *C. sakazakii* isolated from various sources is still scarce and under study. Our current understanding is that *C. sakazakii* infection is due to various virulence factors such as adherence and invasiveness in cell lines [7], gene encoding iron uptake systems, fimbriae, flagella, invasion, outer membranes, serum resistance and spreading, sialic acid utilization, capsule and endotoxin production [15,16,17,18]. In addition, *C. sakazakii* has shown resistance to various antibiotics [19,20], as well as the presence of antibiotic resistance genes, plasmids and mobile genetic elements [21,22].

Studies with whole-genome sequencing (WGS) have shown a high discrimination of the content of conserved and variable genetic information that can discriminate between different species in a more precise way. WGS is used as a tool for identification, typing (multilocus sequence typing (MLST)) core genome multilocus sequence typing ((cgMLST), CRISPR-Cas, serogroup, SNPs), and the detection of genes that confer resistance to antibiotics (ARGs) and/or virulence (VGs), which allows for more precise molecular epidemiological studies [23]. Therefore, the study of complete genomes and their comparison can reveal the role of ARGs and VGs in the pathogenicity of the organism [24].

Therefore, the objective of this study was to use WGS to screen for genes encoding resistance to ARGs, fitness or VGs in *C. sakazakii* isolated from food and the powdered-milk-producing plant environments.

## 2. Results

### 2.1. Sampling and Identification of Isolates

The isolates (n = 15) included in this study (Table 1) were presumptively identified as *Cronobacter* spp. by Matrix Assisted Laser Desorption Ionization—Time of Flight Mass Spectrometry (MALDI-TOF MS). The isolates were then speciated using ribosomal MLST (rMLST; 53 genes). Fourteen strains were identified and *C. sakazakii* and the remainders were identified as *C. dublinensis* (Table 2).

Most *C. sakazakii* were of the neonatal meningitic pathovar sequence type ST4 (60%, 9/15) [5]. The remainder were *C. sakazakii* ST83 (13.3%, 2/15); and single strains of ST1, ST93, ST822, and ST260. The prevalent *C. sakazakii* serotype was O:2 and only the ST1 isolate was serotype O:1. There was no correlation between source of isolation, or serotype, or STs for the *C. sakazakii* ST4 strains (Table 2).

Analysis of the 14 *C. sakazakii* isolates using cgMLST comprising 3678 core genes supported the conventional 7-loci MLST clustering of strains, but also showed that the ST4 strains could be further distinguished (Figure 1).

### 2.2. Antibiotic Resistance Profile

The strains analyzed in this study showed resistance to cephalothin (93%, 14/15), ampicillin (33.3%, 5/15), tetracycline (20%, 3/15), ceftazidime (13.33%, 2/15) and amoxicillin-clavulanic (6.66%, 1/15) (Table 2). Interestingly, only three strains were multidrug-resistant (MDR): 510176-19, 510177-19 and 510182-19. The strains were definied as MDR according to the four families of antibiotics to which these strains were resistant: ampicillin (penicillins), amoxicillin-clavulanic acid (B-lactam combination agents), ceftazidime, cephalothin (cephems) and tetracycline (tetracyclines) (Table 3).

### 2.3. Antibiotic Genes Detected in C. sakazakii Genomes

Twenty genes associated with antibiotic resistance were identified in the 14 *C. sakazakii* genomes. Some of these genes were classified as global regulators, such as HNS and CRP, which were associated with resistance-nodulation-cell division (RND) antibiotic efflux pump AMR families and *rsmA* a global translational regulatory protein RsmA. Other genes were related to antibiotic efflux, reduced antibiotic permeability, target modification or antibiotic inactivation (Appendix A). *C. sakazakii* strains carried the same genes (Figure 2) that were involved in general bacterial porin with reduced permeability to beta-lactams, small multidrug resistance (SMR) antibiotic efflux pump, ATP-binding cassette (ABC) antibiotic efflux pump, resistance-nodulation-cell division (RND), glycopeptide resistance gene cluster, Van ligase, antibiotic-resistant GlpT, and Penicillin-binding protein mutations conferring resistance to beta-lactam antibiotics related to phenotypic profiles (Appendix A).

Three genes (*EF-tu*, *CSA-1* and *qacJ*) were identified in seven, four and two genomes, respectively (Figure 2). The *EF-tu* gene with a *SNP R234F* was detected in 46.7% (7/15) of the genomes. This gene belongs to the AMR gene family elfamycin and their mechanism of resistance is antibiotic target alteration. The gene *CSA-1* was found in 26.7% (4/15) of the genomes. This gene belongs to AMR gene family *CSA beta-lactamases*, conferring resistance to cephalosporin by antibiotic inactivation. Lastly, the *qacJ* gene was identified in 13.3% (2/15) of the genomes of the associated gene family small multidrug-resistance (SMR) antibiotic efflux pump disinfecting agents and antiseptics (Appendix A).

Although most of the genes were identified in all the strains, three genes were absent in the analyzed genomes (Figure 2). The gene *kpnF* was not detected in 60% (9/15) of the genomes. This gene belongs to a major facilitator superfamily (MFS) antibiotic efflux pump. The gene *CSA-2* was not found in 40% (6/15) of analyzed genomes. This gene encodes for a *CSA beta*-lactamase inactivating cephalosporins. The gene *qacG* was not identified in 13.3% (2/15) of genomes. This gene encodes for a small multidrug-resistance (SMR) antibiotic efflux pump associated with resistance to disinfecting agents and antiseptics; it uses the antibiotic efflux as a mechanism of resistance (Appendix A).

The genes *kpnF*, *CSA-1*, and *qacJ* were not identified in the *C. dublinensis* genome (510180-19); however, these genes were detected in most *C. sakazakii* genomes.

### 2.4. Virulence Genes in C. sakazakii Genomes

A total of 99 virulence and fitness genes were identified in the *C. sakazakii* and *C. dublinensis* genomes. The most common genes associated with virulence encode for *ompA*, siderophores and genes involved in metabolism or fitness (Figure 3, Appendix A).

Some genes could be involved in bacterial metabolism, such as: *algG*: bifunctional mannose-1-phosphate guanylyltransferase/mannose-6-phosphate isomerase; *ctpC*: manganese-exporting P-type ATPase CtpC; *galF*: UTP--glucose-1-phosphate uridylyltransferase GalF; *gndA*: NADP+-dependent 6-P-gluconate dehydrogenase; *icl*: isocitrate lyase, *msrA/BpilB*: methionine sulfoxide reductase; *nar*: nitrate reductase; *pchD*: transcriptional regulator.

The virulence genes detected in this study were related to: *icmF1/tssM1, hsiC1/vipB/tssC*, imp: type VI secretion system protein; *hcp*: type VI secretion system receptor/chaperone Hcp; *iucA*: aerobactin synthase, *iucA, iutA*: ferric aerobactin receptor IutA, *entB*: enterobactin synthase component B; *fep*: ferric enterobactin outer membrane transporter; *fes*: ferric enterobactin esterase; *iroN*: siderophore salmochelin receptor IroN, *ompA*: outer membrane protein; *hec*: B family hemolysin secretion/activation protein; *kpsD*: capsule polysaccharide ABC transporter substrate-binding protein; *clvp*: ATP-dependent protease ClpV; *cpsG*: colanic acid biosynthesis phosphomannomutase CpsG; *gacS*: RhoGAP domain-containing protein; *ehaB*: autotransporter adhesin EhaB; and *cdiA*: contact-dependent inhibition effector toxin. The flagella synthesis operon was also identified. Furthermore, plasminogen activator (*cpa*) and utilization of sialic acid (*nanA,K,T*) genes were detected only in the *C. sakazakii* strains and not in *C. dublinensis*. The small heat shock protein sHSP20 was detected in *C. sakazakii* ST4 and ST83, while the locus of heat resistance: *yfdX1GI, yfdX2, hdeDGI, orf11, trxGI, kefB*, and *psiEGI* was only found in the genomes of *C. sakazakii* ST4.

The genes: *bvrS, ehaB, fepB, fliC, fliF, fliJ, hcpC, cpa*, and *nanA,K,T* were not identified in the *C. dublinensis* genome (510180-19); however, these genes were detected in most *C. sakazakii* genomes.

### 2.5. Plasmids and Mobile Genetic Elements in C. sakazakii Genomes

The IncFIB(pCTU3) plasmid was detected in most (8/9) *C. sakazakii* isolates and the ST260 isolate (Table 3). IncFIB(pCTU2), pESA2, rep7a and Col4401 were found in strains 510186-19 and 510180-19, respectively. Regarding the mobile genetic elements (MGEs), the most prevalent were ISEsa1, ISEc52 and ISEhe3 (Table 4).

## 3. Discussion

*C. sakazakii* ST1, ST4 and, to a lesser extent, ST83, are the pathovars that have been most frequently found in PIF marketed in different countries, in PIF production plants and in invasive clinical cases such as fatal meningitis and septicemia [5,25,26,27,28,29]. The other strains, ST93, ST822 and ST260, have not been associated with cases of disease and are of less clinical importance. In this study, we identified nine *C. sakazakii* isolates of ST4 (CC4) serotype *Csak* O:2. These were mostly from environmental sources (5/9) (Table 2), whereas *C. sakazakii* 1 ST1 (CC1) serotype *Csak* O:1 was isolated from food. A previous study carried out on *Cronobacter* in the Americas showed that most of the isolates came from clinical, environmental and infant formula samples from North America (57.4%, n = 465) and Brazil (42.6%, n = 465). In addition, the study reported a total of 75 sequence types, with the most frequent being *C. sakazakii* ST4 (CC4) and ST1 (CC1) [30].

Analysis using a 3678 loci cgMLST scheme revealed a group of eight indistinguisable *C. sakazakii* ST4 isolates from food and the production environment. A previously reported multicenter study assessing the incidence of *C. sakazakii* throughout Europe found that 76.6% (59/77) of the human clinical strains of *Cronobacter* spp. corresponded to *C. sakazakii*. Twelve *C. sakazakii* isolates were ST4, and eight were ST1 [31]. There was only one allele difference between isolates associated with epidemiological outbreaks in Austria in 2016 and 2009, respectively.

*C. dublinensis* has primarily been reported in environmental samples and food ingredients. This is primarily considered an environmental commensal and has rarely been reported in clinical samples [4,32].

In the present study, the *C. sakazakii* strains were resistant to cephalothin, ampicillin tetracycline, ceftazidime and amoxicillin–clavulanic acid. In addition, two strains showed an MDR profile (ampicillin, cephalothin and tetracycline); one was ST 260 (510176-19), and the other was ST4 (510177-19). These two strains were isolated from different foods. Compared to other studies, the resistance observed in this study is low. For example, in Teheran city (Iran), of the 25 *C. sakazakii* strains recovered from PIF, 96% were MDR; these were mainly resistant to amoxicillin–clavulanic acid, amoxicillin, ampicillin, cefoxitin, cefepime, erythromycin, and ceftriaxone, and totally susceptible to trimethoprim/sulfamethoxazole and levofloxacin antibiotics [33]. In China, *C. sakazakii* from PIF and processing environments showed isolates resistant to amoxicillin–clavulanic acid, ampicillin, and cefazolin [34]. MDR isolates have been reported from domestic kitchens in middle Tennessee, (USA); the highest resistance was to penicillin (76.1%), tetracycline (66.6%), ciprofloxacin (57.1%), and nalidixic acid (47.6%) [35]. *C. sakazakii* antibiotic resistance is diverse and depends on the source and geographic location of the strain [33,34,35]. The genomic analysis has enabled the detection of genes involved in antibiotic resistance. The genes are related to cell permeability to beta-lactams and antibiotic efflux pumps. In other studies, *blaCTX* genes are commonly detected in strains resistant to cephalosporins, whereas, in this study, these genes were not detected [36,37]. Another detected gene was *EF-Tu*, which is included in the AMR gene family elfamycin and their mechanism of resistance is antibiotic target alteration. This gene has evolved to be a multifunctional protein in a wide variety of pathogenic bacteria [38]. Functions related to *EF-Tu* vary among microbial species; however, there is a common role of adherence and immune regulation [38]. The *Amp*C *beta*-lactamases designated *CSA-1* and *CSA-2* (from *C. sakazakii*) were found in the 14 *C. sakazakii* strains and have been reported previously [39].

The genomic analysis of the *Cronobacter* strains revealed a number of virulence-related genes. Several putative genes of the Type VI Secretion System cluster were found. The Type VI Secretion System (T6SS) is a protein secretion machinery that transports protein toxins into eukaryotic cells [40,41] and bacteriolytic effectors to target cells [42]. Some genes involved in the iron uptake category (*fur, iroN and PSEEN_RS11615*), responsible for the production of metal binding proteins, were identified in genomic sequences, and are associated with bacterial virulence [43]. The flagella operon was found in all genomes. Flagella has been involved in adherence and proinflammatory response in different pathogens [16]. The most frequently occuring gene, the outer membrane protein A (OmpA), was identified in the genomes. OmpA plays a critical role in host cell invasion [44].

In our research, all 14 *C. sakazakii* strains encoded the *cpa* gene. The *cpa* gene is possibly involved in serum resistance, as well as in the systemic spread of *C. sakazakii*. The isolates harbored the *nanAKT* cassette coding for the use of exogenous sialic acid. Only the *C. sakazakii* species and a few *C. turicensis* strains can catabolize this [45]. Sialic acid is found in gangliosides of the brain, and occurs naturally in breast milk [46]. Powdered infant formula is supplemented with this monosaccharide due to its association with brain development [4]. Sialic acid also regulates the expression of enzymes, such as sialidase and adhesins, or inhibits the transcription factors of the *fimB* gene involved in the adhesion and invasion of epithelial cells [47].

Another important observation was the presence of *rpoS*, *ibpA*, *ibpB* and *clpk* genes in the *C. sakazakii* genomes. These genes are involved in protection against environmental stress [48]. For example, the RpoS regulon, which acts as a transcriptional factor in response to general stress, develops cross-protection against other environmental disturbances, such as the response to oxidative stress and the response to heat stress. In *E. coli*, for example, it represents 10% of genes [49]. On the other hand, the genes encoding for heat shock proteins Hsp15, Hsp20 and HspQ, were found in thermotolerant *C. sakazakii* [50,51]. Therefore, when these environmental *C. sakazakii* are subjected to desiccation or drying processes, as in the preparation of PIF or other foods, they may become more resistant to these stresses, enabling them to persist either in the environments of these production plants or in the final products.

Sixty percent of the *C. sakazakii* strains in our study carried the IncFIB(pCTU3) plasmid, which has been previously reported in environmental sources and food [52,53], as shown in Table 3. The majority of strains carrying the plasmid were *C. sakazakii* ST4. This plasmid is associated with encoding for efflux pumps associated with heavy metals, which could allow the microorganism to adapt to the changing environmental conditions of osmotic stress [54].

In this study, *C. sakazakii* ST4 and ST1 were isolated from environmental strains (Table 2), and harbored genes that encode for antibiotic resistance proteins and virulence factors (Figure 2 and Figure 3). Consequently, there is the potential risk that these isolates could persist in the environment and become a source of food contamination, therefore presenting a risk to the health of consumers.

## 4. Materials and Methods

### 4.1. Sampling

A total of 15 presumptive strains of *Cronobacter* spp. were isolated from 2000 samples from food (n = 997), milk-producing environments (n = 855), and feces from live poultry (n = 148) on the territory of the Czech Republic in 2015. Sampling was performed by instructed veterinarians. Solid, loose and slurry materials were collected in 60–200 mL sterile plastic containers (Dispolab Ltd., Brno, Czech Republic) or in original packaging. Swabs were taken using Transbak system containing Amies broth with active carbon (Dispolab Ltd., Brno, Czech Republic). Environmental swabs, feces and meat samples were stored and transported to the laboratory at +4 °C. Other materials were transported at +21 °C. These strains were analyzed at the bacteriology department, Palacký University Olomouc, Czech Republic (Appendix A).

### 4.2. Isolation and Identification Methods of Cronobacter *spp.*

*Cronobacter* spp. strains were isolated according to the method described by Iversen et al. [55]. Food and environment samples were pre-enriched in buffered peptone water (BPW), followed by *Enterobacteriaceae* enrichment broth (BD Difco, Sparks, MD, USA), then on Brilliance CM 1035 chromogenic agar (Oxoid Thermo-Fisher, UK) and purified on trypticase soy agar (BD Difco, Sparks, MD, USA). Prior to sequencing, the strains were presumptively identified using Matrix Assisted Laser Desorption Ionization—Time of Flight Mass Spectrometry (MALDI-TOF MS) (Bruker, Billerica, MA, USA) and with the MALDI Biotyper Compass IVD 4.1.60 software (Bruker, Billerica, MA, USA) described by Lepuschitz et al. [56]. The identification of the *Cronobacter* spp. strains was confirmed with the Ribosomal Multilocus Sequence Typing (rMLST) software available at https://pubmlst.org/species-id (accessed on 2 February 2023) [57].

### 4.3. Whole-Genome Sequencing (WGS)

*Cronobacter* spp. isolates were cultured in Columbia blood agar plates (bioMérieux, Marcy-l’Étoile, France) at 37 °C for 24 h. For WGS, DNA was isolated from bacterial cultures with the MagAttract HMW DNA Kit (Qiagen, Hilden, Germany) according to the manufacturer’s instructions. The quantification of input DNA was performed with a Qubit 2.0 fluorometer (Thermo Fisher Scientific, Waltham, MA, USA) and the double-stranded DNA (dsDNA) BR assay kit (Thermo Fisher Scientific). Nextera XT chemistry (Illumina Inc., San Diego, CA, USA) was used to prepare sequencing libraries for a 2 × 300 bp paired-end sequencing run on an Illumina MiSeq sequencer. Samples were sequenced to achieve a minimum of 80-fold coverage using standard Illumina protocols. The resulting FASTQ files were quality-trimmed and de novo assembled with the SPAdes v3.11.1 software. Contigs were filtered for a minimum of 5-fold coverage and 200  bp minimum length with the Ridom SeqSphere+ software v8.3 (Ridom, Münster, Germany) [31]. Raw reads were quality-controlled using FastQC v0.11.9. Trimmomatic v0.36 [58] was used to remove adapter sequences and to trim the last 10 bp of each sequence and sequences with quality scores  < 20. Reads were assembled using SPAdes v3.11.1 [59]. Contigs were filtered for a minimum coverage of 5× and a minimum length of 200 bp using SeqSphere+ software v8.3.0 (Ridom GmbH, Würzburg, Germany) [58] (Appendix A).

### 4.4. Sequence Type (ST) and Core Genome Multilocus Sequence Typing (cgMLST) of Cronobacter sakazakii

A total of 3678 targets were used for core genome multilocus sequence typing (cgMLST) scheme of *Cronobacter* spp., with strain ATCC BAA-894 as a reference using a target gene loci task template of the Ridom SeqSphere+ software v8.3.0 (Ridom, Münster, Germany) [31,60]. According to this cgMLST scheme, isolates were visualized with a minimum spanning tree (MST) to establish their genotypic relationships and define those isolates with maximum differences of 10 alleles as clusters [31]. In addition, the sequences of the seven housekeeping genes of the conventional MLST for *C. sakazakii* (*atpD, fusA, glnS, gltB, gyrB, infB* and *ppsA*) were extracted and cross-checked against the *Cronobacter* MLST database https://pubmlst.org/organisms/cronobacter-spp/ (accessed on 2 February 2023) [61].

### 4.5. Determination of Serotypes

The profiles of *gnd* and *galF* genes specific to the *Cronobacter* spp. serotype O region were determined by WGS sequence analysis with the BIGSdb tool available in the PubMLST database (http://pubmlst.org/cronobacter/) (accessed on 2 February 2023) [18].

### 4.6. Antibiotic Susceptibility

To assess antibiotic susceptibility, the disk diffusion method was used, in accordance with the recommendations of the Clinical and Laboratory Standards Institute [62]. The commercial disks that were used consist of ampicillin (10 μg), amoxicillin–clavulanic acid (20/10 µg), ceftazidime (30 µg), ciprofloxacin (5 μg), chloramphenicol (30 μg), cefotaxime (30 μg), gentamicin (10 μg), cephalothin (30 μg), tetracycline (30 µg) and nalidixic acid (30 µg). The characterization of the resistance/susceptibility profiles was determined according to the manufacturer’s instructions. The *Escherichia coli* ATCC 25922 and *Pseudomonas aeruginosa* ATCC 27853 strains were used as internal control.

### 4.7. Detection of Antibiotic Resistance and Virulence Genes

The existence of virulence genes was confirmed by ResFinder tool from the Center of Genomic Epidemiology (CGE) (http://www.genomicepidemiology.org) (accessed on 5 March 2023) [63]. Thresholds for the target scanning procedure were set with a required identity of ≥90% to the reference sequence and an aligned reference sequence ≥99%. The Comprehensive Antibiotic Resistance Database (CARD) with the “perfect” and “strict” default settings for sequence analysis [64], the Task Template AMRFinderPlus 3.2.3 available in Ridom SeqSphere+ v7.8.0 software using the EXACT method at 100%, and BLAST alignment for protein identification available in the AMRFinderPlus database were used for antimicrobial resistance genes. For the search for genes and proteins associated with virulence and fitness, the Virulence Factor Database (VFDB), available at http://www.mgc.ac.cn/VFs/main.htm (accessed on 5 March 2023), was downloaded and compared with each of the 15 genomes using the blast tool ncbi-blast-2.13.0+-x64-arm-linux.tar.gz, available at https://ftp.ncbi.nlm.nih.gov/blast/executables/blast+/LAST/ (accessed on 5 March 2023) on command line.

### 4.8. Detection of Plasmids and Mobile Genetic Elements (MGEs)

The PlasmidFinder v2.1 and MobileElementFinder v1.0 tools were used to detect plasmids and mobile genetic elements (MGEs), respectively. The selected minimum identity was 95% and 90%, respectively (http://www.genomicepidemiology.org/) (accessed on 5 March 2023) [65,66].

## 5. Conclusions

The *C. sakazakii* isolates analyzed in this study harbored ARGs and VGs, which contributed to their persistence in powdered-milk-producing environments, and increased the risk of infection in susceptible population groups due to the presence of multi-resistant strains to antibiotics, reducing the possibility of treating of the infection [67,68].

## Figures and Tables

**Figure 1 antibiotics-12-00851-f001:**
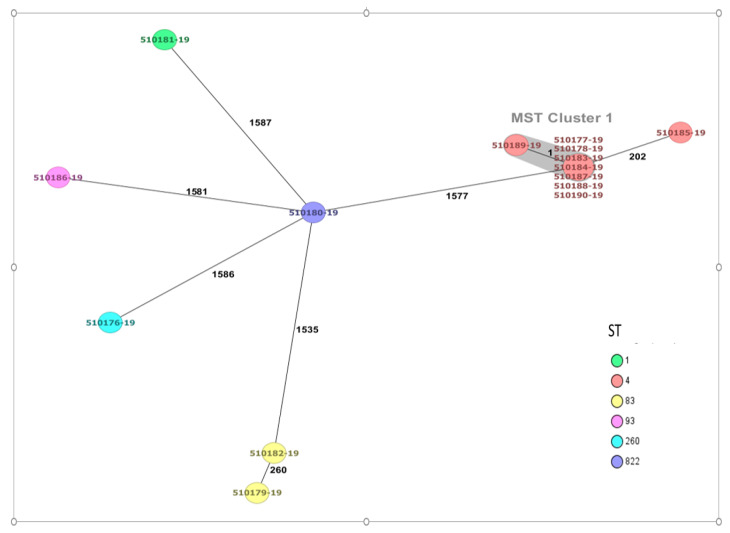
Minimum spanning tree (MST) of fourteen *C. sakazakii* strains. The isolates are represented as colored circles according to their sequence type (ST) as defined using the 7-loci MLST scheme (STs). Black numbers on the connection lines indicate the number of allelic differences between isolates from the cgMLST scheme comprising 3678 target genes for *C. sakazakii*. Isolates falling under the cluster threshold of 10 alleles are marked in grey as clusters.

**Figure 2 antibiotics-12-00851-f002:**
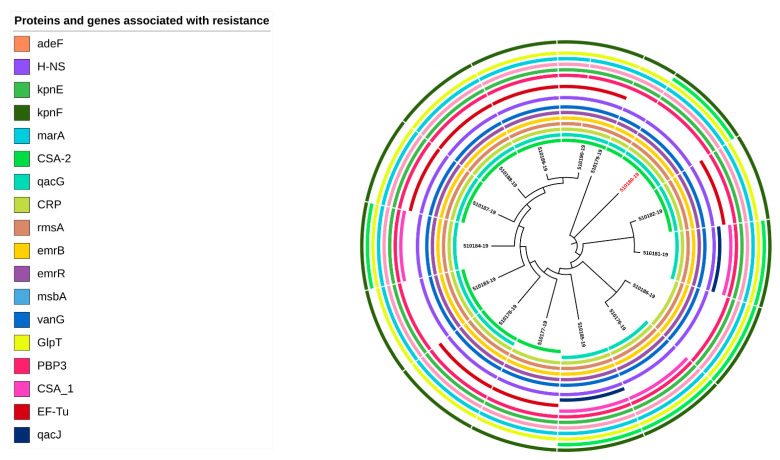
Antibiotic-resistant genes of fifteen *Cronobacter* spp. strains identified by Comprehensive Antibiotic Resistance Database (CARD). *C. dublinensis* genome is shown in red color. The visualization was performed with the iTol program.

**Figure 3 antibiotics-12-00851-f003:**
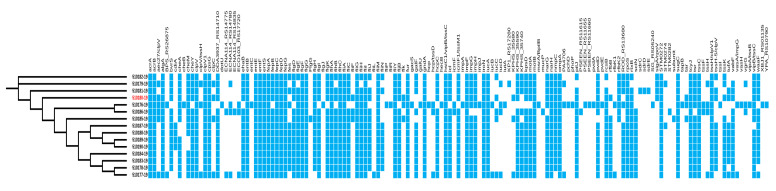
Virulence genes identified in *C. sakazakii* and *C. dublinensis* genomes (red color font). The visualization was performed with the iTol program.

**Table 1 antibiotics-12-00851-t001:** Numbers and types of examined samples in the study.

Group of Samples	Specific Commodity	Number of Examined Samples
Swabs from the food production environment	Swabs	855
Poultry	Feaces	148
Food	Chockolate	72
	Caramel	55
	Spice (dill, pepper)	260
	Poultry meat	77
	Dried cow milk	244
	Wheat flour	91
	Seeds (soy, pistachio, barley, mustard)	198
Total		2000

**Table 2 antibiotics-12-00851-t002:** Identification of *Cronobacter* spp. strains isolated from different sources by MALDI-TOF MS and whole-genome sequencing rMLST.

Sample ID	* PubMLST ID	Source	WGS rMLST Result	ST	CC	Serotype (*gnd-galF* Alleles)	Collection Date
510177-19	3823	Food	*Cronobacter sakazakii*	4	4	*Csak* O:2	2015
510178-19	3824	Food	*Cronobacter sakazakii*	4	4	*Csak* O:2	2015
510183-19	3828	Production environment	*Cronobacter sakazakii*	4	4	*Csak* O:2	2015
510184-19	3829	Production environment	*Cronobacter sakazakii*	4	4	*Csak* O:2	2015
510185-19	3830	Food	*Cronobacter sakazakii*	4	4	*Csak* O:2	2015
510187-19	3831	Food	*Cronobacter sakazakii*	4	4	*Csak* O:2	2015
510188-19	3832	Production environment	*Cronobacter sakazakii*	4	4	*Csak* O:2	2015
510189-19	3833	Production environment	*Cronobacter sakazakii*	4	4	*Csak* O:2	2015
510190-19	3834	Production environment	*Cronobacter sakazakii*	4	4	*Csak* O:2	2015
510181-19	3826	Food	*Cronobacter sakazakii*	1	1	Csak O:1	2015
510179-19	3825	Production environment	*Cronobacter sakazakii*	83	83	Csak O:7	2015
510182-19	3827	Food	*Cronobacter sakazakii*	83	83	*Csak* O:7	2015
510176-19	3695	Food	*Cronobacter sakazakii*	260	-	*Csak* O:1	2015
510186-19	3697	Hen	*Cronobacter sakazakii*	93	-	*Csak* O:7	2015
510180-19	3696	Food	*Cronobacter dublinensis*	822	-	ND	2015

* MLST database ID; WGS rMLST: Whole-Genome Sequencing ribosomal MLST; ST: sequence type; CC: clonal complex; -: no associated clonal complex; ND: not determined.

**Table 3 antibiotics-12-00851-t003:** Antibiotic resistance profile of *Cronobacter* spp. strains.

ST	Strains ^a^	AM (10 µg)	AMC (20/10 µg)	CAZ (30 µg)	CIP (5 µg)	CL (30 µg)	CTX (30 µg)	GE (10 µg)	KF (30 µg)	TE (30 µg)	W (30 µg)
1	510181-19	S	S	S	S	S	S	S	R	S	S
4	510177-19	R	S	S	S	S	S	S	R	R	S
4	510178-19	S	S	S	S	S	S	S	R	S	S
4	510183-19	S	R	S	S	S	S	S	R	S	S
4	510184-19	S	S	S	S	S	S	S	R	R	S
4	510185-19	S	S	S	S	S	S	S	R	S	S
4	510187-19	S	S	S	S	S	S	S	R	S	S
4	510188-19	R	S	S	S	S	S	S	R	S	S
4	510189-19	S	S	S	S	S	S	S	R	S	S
4	510190-19	S	S	S	S	S	S	S	R	S	S
83	510179-19	R	S	S	S	S	S	S	R	S	S
83	510182-19	R	S	R	S	S	S	S	R	S	S
260	510176-19	R	S	S	S	S	S	S	R	R	S
822	510180-19 *	S	S	S	S	S	S	S	S	S	S

^a^ All strains were *C. sakazakii* except for 510180-19; * which was *C. dublinensis*. AM: ampicillin; AMC: amoxicillin-clavulanic acid; CAZ: Ceftazidime; CIP: ciprofloxacin; CL: chloramphenicol; CTX: cefotaxime; GE: gentamicin; KF: cephalothin; TE: tetracycline; W: nalidixic acid; R: Resistance; S: Susceptibility.

**Table 4 antibiotics-12-00851-t004:** Plasmids and mobile genetic elements of *Cronobacter* spp. strains.

ID Strain ^a^	ST	Plasmid	Plasmid Accession Number	Mobile Genetic Elements
510177-19	4	IncFIB(pCTU3)	FN543096	ISEhe3, ISEsa1, ISEc52
510178-19	4	IncFIB(pCTU3)	FN543096	ISEhe3, ISEsa1, ISEc52
510183-19	4	IncFIB(pCTU3)	FN543096	ISEc52
510184-19	4	IncFIB(pCTU3)	FN543096	ISEhe3, ISEsa1
510185-19	4	-----	-----	ISEhe3
510187-19	4	IncFIB(pCTU3)	FN543096	ISEhe3, ISEsa1, IS26, ISEc52
510188-19	4	IncFIB(pCTU3)	FN543096	ISEsa1,IS26, ISEc52
510189-19	4	IncFIB(pCTU3)	FN543096	ISEhe3, ISEsa1
510190-19	4	IncFIB(pCTU3)	FN543096	ISEhe3, ISEsa1, ISEc52
510179-19	83	rep7a	SAU83488	-----
510180-19	822	IncFII(pCTU2) pESA2	FN543095 CP000784	ISEch12
510186-19	93	Col440l	Cp023920.1	ISEsa1, ISKpn34
510176-19	260	IncFIB(pCTU3)	FN543096	ISEsa1, IS26, cn_6897_IS26

^a^ All strains were *C. sakazakii* except for 510180-19 which was *C. dublinensis*.

## Data Availability

The *Cronobacter sakazakii* and *Cronobacter dublinensis* isolates were submitted to https://pubmlst.org/organisms/cronobacter-spp with ID 3695-3697 and 3823-3834.

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
