# Peer review of "Screening of Antibiotic and Virulence Genes from Whole Genome Sequenced Cronobacter sakazakii Isolated from Food and Milk-Producing Environments"

_antibiotics, 2023, doi:10.3390/antibiotics12050851_

Round 1
Reviewer 1 Report
The authors presented a manuscript entitled "Screening of antibiotic and virulence genes from Cronobacter sakazakii isolated from food and milk-producing environ ments using WGS". They propose using WGS to screen for antimicrobial resistance genes, and Virulence genes.
The manuscript is well written but English can be improved. In general, is interesting data for the field, I have some observations/suggestions below:
Minor
L 28, 203, 311, 318. Please shift "Cronobacter" to "Cronobacter", please check the entire manuscript in order to all the mentions of the genera, species, and genes must be italicized.
L41. Please erase the word "it" after "spp" as well as the final period in all the "spp." you used in the manuscript and tables.
L166. Add a ":" after "algG" to homogeneize the text
L172. Shift "icmF1/tssM1" to italicized, you are referring to a genes
L 208. Please add a period after "isolates".
L291 Please check "Croobacter"?
L331Please check the paragraph shape, there is a n extra "enter"
Major
Table 3. Why is necessary to mention strains 510181-19 & 510182-19 if none of them have plasmids or MGE identified?
You worked with 15 strains, do your results based on this little "n" strong enough to support the conclusions?
The conclusions do not indicate whether the objective was achieved or not.
Author Response
Point 1: The manuscript is well written but English can be improved. In general, is interesting data for the field, I have some observations/suggestions below:
Response 1: English was completely revised by Prof. Stephen Forsythe, native speaker
Point 2: L 28, 203, 311, 318. Please shift "Cronobacter" to "Cronobacter", please check the entire manuscript in order to all the mentions of the genera, species, and genes must be italicized.
Response 2: The corrections were made
Point 3: L41. Please erase the word "it" after "spp" as well as the final period in all the "spp." you used in the manuscript and tables.
Response 3: The corrections were made
Point 4: L166. Add a ":" after "algG" to homogeneize the text
Response 4: The corrections were made
Point 5: L172. Shift "icmF1/tssM1" to italicized, you are referring to a genes
Response 5: “icmF1/tssM1" was italicized
Point 6: L 208. Please add a period after "isolates".
Response 6: The corrections were made
Point 7: L291 Please check "Croobacter"?
Response 7: The corrections were made: Cronobacter
Point 8: L331Please check the paragraph shape, there is a n extra "enter"
Response 8: Corrected in the text
Major
Point 9: Table 3. Why is necessary to mention strains 510181-19 & 510182-19 if none of them have plasmids or MGE identified?
Response 9: The table was corrected and deleted the strains 510181-19 and 510182-19
Point 10: You worked with 15 strains, do your results based on this little "n" strong enough to support the conclusions?
Response 10: The 15 strains were isolated from 2000 samples, therefore, we consider that the results support our conclusion.
Point 11: The conclusions do not indicate whether the objective was achieved or not.
Response 11: The conclusion was improved according to the objective
Reviewer 2 Report
Attached

Author Response
Point 1: The manuscript presents the pertinent issues on regards to the genomic features and phenotypic antimicrobial resistance of Cronobacter spp. isolated from the food, hen and food producing environment. The authors showed the clinical importance of this bacteria and group of people more likely to acquire infections. However, the introduction and discussion sections are lacking strain specific clinical importance in case there is any variation in clinical presentation among different strains based on available literature. Since the study involved detection ARGs and phenotypic antimicrobial resistance,
I expected the authors to account on any relationship on ARGs detected and phenotypic antimicrobial resistance pattern observed. In whole genome sequencing the examination of the quality indices of the sequence generated prior continuing with further analysis is extremely important, however, the current study did not account on this. Finally, the conclusion was more based the literature rather than the result generated by this study. The authors are insisted to concentrate on the study results to draw the conclusion before embarking on the available literature.
Response 1: The manuscript was revised again and the introduction and conclusions were improved according to what was commented by the reviewer. In addition, we reordered the information in the tables for better understanding.
Regarding the quality of the sequences, parameters were added in the text (section 4.3) and Table 3 was attached in supplementary material with all the information requested.
Point 2: Line 97: What do you mean by “not determined” determination was not successful or was not attempted?
Response 2: We have revised this Table with the serotype for the C. sakazakii strains. However, although the O-antigen biosynthesis flanking genes galF and gnd have been determined for the single C. dublinensis strain (ID3696), these do not correspond with any described C dublinensis serotype, and therefore remains ‘unknown’, until further whole genome sequencing is undertaken – which was considered outside the scope of the current study.
For the Table:
ID 3825, 3827, & 3697 : Csak O:7
ID 3695 Csak O:1
ID 3696 Unknown, cannot be determined
Point 3: Line 125. “…….classified as global regulators.” Describe more on this
Response 3: Added: HNS and CRP, that were associated to resistance-nodulation-cell division (RND) antibiotic efflux pump AMR families
Point 4: Table 1 One sample was isolated in the hen, was it live, dead or slaughtered for consumption? How many hens were sampled in total? You are indicating other strains to be isolated in food, what type of food? There is a need of segregating those samples (2000) to show the sources for better risk characterization
Response 4: A total of 15 presumptive strains of Cronobacter spp. were isolated from 2,000 samples from food (n=997), environment related to food production (n=855) and feaces from live poultry (n= 148) on the territory of the Czech Republic in 2015. The exact numbers and sample commodities are shown in the table. Sampling was performed by instructed veterinarians. Solid, loose and slurry materials were sterile collected in 60 – 200 ml plastic containers (Dispolab Ltd. Brno, Czech Republic) or in original package. Swabs were taken using Transbak system containing Amies broth with active carbon (Dispolab Ltd. Brno, Czech Republic). Environmental swabs, feces and meat samples were stored and transported to the laboratory at +4°C. Other materials were transported at +21°C. These strains were analyzed at the bacteriology department, Palacký University Olomouc, Czech Republic.
|
Group of samples |
Specific commodity |
Number of examined samples |
|
Swabs from the food production environment |
Swabs |
855 |
|
Poultry |
Feaces |
148 |
|
Food |
Chockolate |
72 |
|
Caramel |
55 |
|
|
Spice (dill, pepper) |
260 |
|
|
Poultry meat |
77 |
|
|
Dried cow milk |
244 |
|
|
Wheat flour |
91 |
|
|
Seeds (soy, pistachio, barley, mustard) |
198 |
|
|
Total |
|
2000 |
Point 5: Line 217. “………..two strains showed a MDR profile…..” For which drugs?
Response 5: The sentence was modified: Interestingly, only two strains were multidrug resistant (MDR), resistant to three or more antibiotic families. The MDR strains 510176-19 and 510177-19 were resistant to: ampicillin (penicillins), cephalothin (cephems) and tetracycline (tetracyclines)
Point 6: Line 217. “………from different food………” What type of food?
Response 6: The sentence was modified.
Point 7: Line 223-224. “In contrast that has been reported in USA and China” Thus sentence has no connection with the previous sentence, contrast in which sense?
Response 7: The sentence was modified.
Point 8: Line 233. “………..Finding that the 96% of strains were MDR.”I think you need to re-write this section of the sentence or create a new sentence.
Response 8: The section was rewritten.
Point 9: Line 277. How samples were collected, what was your sampling strategy? Any sample size determination approach? How many sample were from environment, food and hen?
Response 9: The section was rewritten.
Point 10: Line 294. More sequence quality parameters is required in this section (Average coverage depth, contig number, size of the contig, mean of the N50 etc.).
Response 10: Regarding the quality of the sequences, parameters were added in the text (section 4.3) and Table 3 was attached in supplementary material with all the information requested.
Point 11: Line 357-358 “Considering the severity of infections associated with C. sakazakii ST4 (CC4) and ST1 358 (CC1) in the susceptible population” Where are you referring this? Use the results generated by your study to draw conclusion
Response 11: The section was rewritten.
Point 12: Line 100-101 “A correlation between source of isolation, serogroup, or sequence type, was did not find.”. Not clear
Response 12: The sentence was modified.
Point 13: Line 357-363 Conclusion was not drawn from the results generated by the study
Response 13: The conclusion was rewritten to represent the results of the study
Reviewer 3 Report
Dear authors,
See attached for my comments and revision recommendations.
Reviewer.

Author Response
Point 1: Line 23 Spell it out since it's mentioned for the first time.
Response 1: Add Cronobacter
Point 2: Line 25 Add antibiotic between and and resistance
Response 2: Add
Point 3: Line 56 PIF?
Response 3: Yes, PIF and was corrected
Point 4: Line 72 Spell out each of these acronyms.
Response 4: Done
Point 5: Line 74 Rephrase to improve information clarity.
Response 5: The phrase was rewritten for clarity
Point 6: Line 83 Change it to identified.
Response 6: Changed
Point 7: Line 94 Rephrase; you mentioned in 2.1 that you used WGS rMLST to confirm Cronobacter sp.
Response 7: the rMLST tool is used with the WGS data
Point 8: Line 97: Change it to determined.
Response 8: Was changed
Point 9: Line 100: Change it to serotype.
Response 9: Was changed
Point 10: Line 101: Rewording requested.
Response 10: Rewording done
Point 11: Line 108-110. Reformat request: Align Left.
Response 11: Aligned to left
Point 12: Line 114. Correction request: six (check out table 2 for strains that possess 2 or > 2 MDR)
Response 12: Was corrected
Point 13: Line 124-132.Rephrase this paragraph to avoid text redundancy.
Response 13: The phase was rewritten
Point 14: Line 134. Clarify which other genes.
Response 14: Add to the between belongs and AMR.
Point 15: Line 136. Add to the between belongs and AMR.
Response 15: Done
Point 16: Line 138: Add to the between belongs and AMR.
Response 16: Add “to”
Point 17: Line 146. Add this statement of the analyzed genomes between (6/15) and ,.
Response 17: The reviewer’s request is very unclear. I think it is just editorial and does not the science of the sentence. My best guess is that the review just wants the sentence changed to ‘was not found in 40% (6/15) of the analysed genomes.’
Point 18: Line 146-149 Reorganize the sentences.
Response 18: Done
Point 19: Line 162. Clarification request: Are these 99 genes found in each of 15 strains you reported herein?
Response 19: Yes, in all strains
Point 20: Line 163. Rewrite these paragraphs to avoid text redundancy.
Response 20: The phase was rewritten
Point 21: Line 163. Replace of with with.
Response 21: Changed
Point 22: Line 164: Edit: Change encoded to encode
Response 22: Was changed
Point 23: 166: Edit: mannose-1-phosphate
Response 23: Was edited
Point 24: Line 180. Image improvement request to enhance text visibility.
Response 24: The image was submitted with a quality of 300 dpi
Point 25: Line 191. What about rep7a and pESA2 plasmids???
Response 25: Rep7a and pESA2 were added
Point 26: Line 198. Use C.
Response 26: Changed
Point 27: Line 200: What about the remaining three strains???
Response 27: Insert ~line 203-204, after [5,25-29]: ’The other strains ST93, ST822 , and ST260 have not been associated with cases of disease and are of less clinical importance.’
Point 28: Line 201-214. Rephrase to improve information clarity.
Response 28: The wording has been improved
Point 29: Line 213: Explain what this information means.
Response 29: The wording has been improved
Point 30: Line 215: Correct the citation bracket.
Response 30: The citation was corrected
Point 31: Line 217: isolated
Response 31: The wording has been improved
Point 32: Line 220: There are more (six) MDR strains (check out Table 2).
Response 32: Only two strains showed that same resistance profile: ampicillin, cephalothin and tetracycline
Point 33: Line 222. Information disconnection; reorganize the information.
Response 33: The wording has been improved
Point 34: Line 224: Need reference(s) or, if this is your experimental findings, provide figure/table #.
Response 34: The wording has been improved
Point 35: Line 225: Not clear what them refers to; specify them.
Response 35: The wording has been improved
Point 36: Line 227.Not.
Response 36: The wording has been improved
Point 37: Line 233: Need explanation why novel and/or reference(s).
Response 37: The wording has been improved
Point 38: Line 245: Need explanation and/or reference(s) why it's expected.
Response 38: The wording has been improved and add a new reference
Point 39: Line 247-255. Rephrase this paragraph to improve information smoothness.
Response 39: The wording has been improved
Point 40: Line 257. Add reference(s).
Response 40: Added a new reference
Point 41: Line 259-261: Rephrase to avoid text redundancy.
Response 41: The wording has been improved
Point 42: Line 262-266 Rephrase to improve information clarity.
Response 42: The wording has been improved
Point 43: Line 268-270: You need to mention this information in your results. And, it's not clear the connection between efflux pump and osmotic stress; provide explanation.
Response 43: The wording has been improved
Point 44: Line 272-274 Mention which of your data infer that; you need to list the genes out and specify your result Table/Figure #.
Response 44: Insert ‘As shown in Table 3’ [51,52]
Point 45: Line 278: Rewrite this words as milk-producing environment.
Response 45: Corrected
Point 46: Line 282: Mention the analysis method used.
Response 46: Add “method”
Point 47: Line 286: Correction request.
Response 48: The correction was made
Point 49: Line 288: Spell out.
Response 49: The correction was made
Point 50: Line 291: Correction request.
Response 50: The correction was made
Point 51: Line 310-318 Rephrase this paragraph to improve information clarity.
Response 51: : The paragraph has been improved
Point 52: Line 331: Move this paragraph up to line 331.
Response 52: The correction was made
Point 53: Line 338: Add reference(s).
Response 53: Added a new reference
Point 54: Line 352: Add , respectively between (MGEs) and ..
Response 54: Added respectively
Point 55: Line 359-362: Rephrase to improve information clarity.
Response 55: The phrase has been improved
Round 2
Reviewer 1 Report
None
Author Response
Point 1: Reviewer 1 comments: None
Response 1: Thanks for your comments
Reviewer 2 Report
Including the table showing the source of the samples in the manuscript not in the reviewer response document only
Author Response
Point 1: Reviewer 2 comments: Including the table showing the source of the samples in the manuscript not in the reviewer response document only
Response Point 1: Table 1 is added in the manuscript
Reviewer 3 Report
Dear authors,
Revision recommendations, including the previous revision requests, are in attached.
Reviewer.

Author Response
Point 1: Line 25: Replace with antibiotic resistance
Response Point 1: The sentence was modified.
Point 2: Line 95: Rephrase; you mentioned in 2.1 that you used WGS rMLST to confirm Cronobacter sp
Response Point 2: rMLST is added
Point 3: Line 118: If three-drug resistance is your MDR definition, you need to mention that in the sentence.
Response Point 3: The sentence was modified.
Point 4: Line 124 Table 2: rearrange the strains in numerical sequence or in ST sequence.
Response Point 4: Strains ordered by ST for clarity
Point 5: Line 129-132: List the number of global regulating genes.
Response Point Description of the last global regulator gene was added.
Point 6: Line 134-139: Rephrase this sentence to improve clarity.
Response Point 6: The sentence was rephrased.
Point 7: Line 161: Tell the readers if this 99 virulence and fitness genes are present in the C. Sakazakii strains analyzed and C. dublinensis.
Response Point 7: Added a new paragraph for clarity
The genes: bvrS, ehaB, fepB, fliC, fliF, fliJ, hcpC, cpa, and nanA,K,T were no-identified in the C. dublinensis genome (510180-19); however, these genes were detected in most C. sakazakii genomes.
Point 8: Line 190: Format: Align left
Response Point 8: left aligned format
Point 9: Line 206-207: Edit the sentence: nine C. sakazakii isolates of ST4 (CC4) serotype?!
Response Point 9: serotype Csak O:2 is added
Point 10: Line 207: Include (Table 1) here.
Response Point 10: Table 2 was included (previous Table 1)
Point 11: Line 214-217: Provide a reference.
Response Point 11: Reference 31 is relocated
Point 12: Line 223: Inconsistent spelling. Spell check.
Response Point 12: The sentence was rephrased.
Point 13: Line 224: Inconsistent spelling. Spell check.
Response Point 13: The sentence was rephrased.
Point 14: Line 224: cephalothin
Response Point 14: The mistake was corrected.
Point 15: Line 226: References needed.
Response Point 15: Reference is added
Point 16: Line 229: Remove in contrast, as the information in previous sentence doesn't contradict with this sentence.
Response Point 16: In contrast was removed
Point 17: Line 226-234: Consider reorder/rearrange the information in lines 226-234.
Response Point 17: The information was rearranged
Point 18: Line 231: Revise as this words do not make sense.
Response Point 18: The sentence was modified
Point 19: Line 230: Provide reference(s).
Response Point 19: References was provided
Point 20: Line 236: Provide reference(s).
Response Point 20: Pakbin B, Brück WM, Allahyari S, Rossen JWA, Mahmoudi R. Antibiotic Resistance and Molecular Characterization of Cronobacter sakazakii Strains Isolated from Powdered Infant Formula Milk. Foods. 2022, 11, 1093.
Point 21: Line 237-238: Provide reference(s).
Response Point 21: References was provided
Point 22: Line 240: Rephrase.
Response Point 22: The sentence was rephrased
Point 23: Line 240-241: Provide reference(s).
Response Point 23: References was provided
Point 24: Line 244-245: They aren't novel anymore since they have been studied previously. Rephrase.
Response Point 24: The sentence was modified
Point 25: Line 255-256: Explain why you come to conclusion that this outer membrane protein-expressing gene is found in the bacterium genome.
Response Point 25: The sentence was modified.
Point 26: Line 283-286: You need to direct reader to your figure or table data regarding this paragraph.
Response Point 26: Added Table and Figures
Point 27: Line 297: Spacing request.
Response Point 27: added spacing
Point 28: Line 301: Name the method used.
Response Point 28: Added Cronobacter spp methods
Point 29: Line 315: Why blood agar was used?
Response Point 29: It´s just our standard agar we use - we are clinical microbiology lab - and we know that taking bacteria from this agar works fine for subsequent DNA isolation and NGS
Point 30: Line 316: Spacing requested previously.
Response Point 30: added spacing
Point 31: Line 380-382: Elaborate why ARG and VG could increase bacterium persistence and infection. Provide reference(s).
Response Point 31: The sentence was modified and added 2 references